# BlockCert: Certified Blockwise Extraction of Transformer Mechanisms

## Abstract

Mechanistic interpretability aspires to reverse-engineer neural networks into explicit algorithms, while model editing seeks to modify specific behaviours without retraining. Both areas are typically evaluated with informal evidence and ad-hoc experiments, with few explicit guarantees about how far an extracted or edited model can drift from the original on relevant inputs. We introduce BLOCKCERT, a tooling pipeline for *certified blockwise extraction* of transformer mechanisms, and outline how a lightweight extension can support *certified local edits*. Given a pre-trained transformer and a prompt distribution, BLOCKCERT extracts structured surrogate implementations for residual blocks together with machine-checkable certificates that bound approximation error, record coverage metrics, and hash the underlying artifacts. We formalize a simple Lipschitz-based composition theorem in Lean 4 that lifts these local guarantees to a global deviation bound. Empirically, we apply the framework to GPT-2 small, TinyLlama-1.1B-Chat, Llama-3.2-3B, and Llama-2-7B. Across these models we obtain high per-block coverage and small residual errors on the evaluated prompts, and in the TinyLlama setting we show that a fully stitched model matches the baseline perplexity within $\approx 6 \times 10^{-5}$ on stress prompts. Our results suggest that blockwise extraction with explicit certificates is feasible for real transformer language models and offers a practical bridge between mechanistic interpretability and formal reasoning about model behaviour.

## 1 Introduction

Large language models (LLMs) have rapidly become core infrastructure for scientific research, software engineering, and high-stakes decision support. At the same time, we still lack robust tools for understanding, validating, and safely modifying their internal mechanisms. Mechanistic interpretability aims to address this by reverse-engineering neural networks into human-understandable components and circuits (Olah et al., 2020; Elhage et al., 2022; Nanda et al., 2023), but existing work is typically evaluated with bespoke visualizations or small-scale experiments, without explicit, machine-checkable guarantees.

In parallel, the formal-methods community has developed powerful tools for proving properties of neural networks (Katz et al., 2017; 2019; Wang et al., 2021), and the programming-languages community has shown how to ship code with machine-checkable proofs of safety (*proof-carrying code*) (Necula, 1997). However, these lines of research have had limited impact on day-to-day interpretability practice. Verification tools often do not scale to modern LLMs, and proof obligations are difficult to relate to the informal, circuit-level stories that interpretability researchers actually use.

**Goal.** We would like a middle ground between fully formal verification and informal interpretability stories: a workflow in which reverse-engineered mechanisms are accompanied by *explicit certificates* that (i) precisely specify what has been extracted, (ii) quantify how closely the extract matches the original network on concrete data, and (iii) can be automatically checked by any third party with access to the artifacts. Ideally, these local certificates can then be composed to reason about global model behavior.

**This paper.** We propose BLOCKCERT, a framework for *certified blockwise extraction* of transformer mechanisms. Given a pre-trained transformer, a set of prompts, and access to intermediate activations, BLOCKCERT produces, for each residual block:

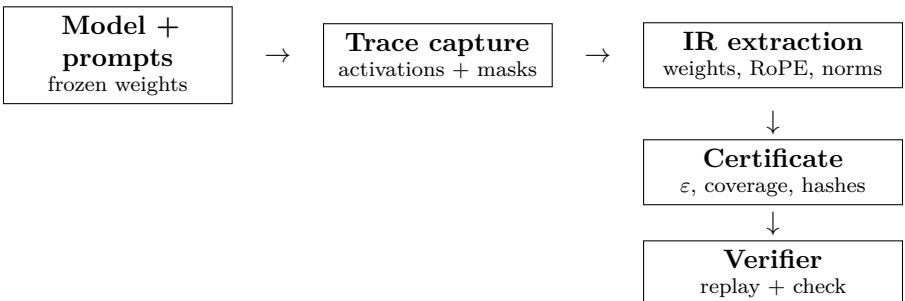

Figure 1: High-level BLOCKCERT/BLOCKCERT-EDIT workflow. Given a pre-trained transformer and a prompt distribution, we record block-level traces, construct IR surrogates with local error and coverage metrics, and package these into machine-checkable certificates. BLOCKCERT-EDIT applies simple local edits (e.g., scaling or swapping mechanisms) and produces analogous edit certificates on the same prompt distribution.

1. a structured surrogate implementation $\hat{B}_i$ in an explicit intermediate representation, and
2. a JSON certificate describing the data distribution used for extraction, the achieved approximation error and coverage metrics, and cryptographic hashes of the associated artifacts (weights, probes, masks).

An independent verifier can re-load the artifacts, recompute the metrics, and check that the hashes match. We also provide a composition mechanism that aggregates per-block certificates into a *full-model certificate* summarizing stitched-model replay metrics. We view BLOCKCERT primarily as a tooling and standardization effort: the contribution is a concrete pipeline, certificate format, and verification workflow that make block-level artifacts reproducible and auditable, rather than a new interpretability algorithm.

We further formalize a simple global bound: if each baseline/stitched block pair $(B_i, \hat{B}_i)$ is locally $\varepsilon_i$-sound and the blocks are $L_i$-Lipschitz, then the composed stitched model is globally bounded by a Lipschitz-weighted sum of the $\varepsilon_i$, which specializes to a $\sum_i \varepsilon_i$ bound when $L_i \leq 1$. This theorem is mechanized in Lean 4 and instantiated for TinyLlama using empirical per-block error bounds.

**Why blockwise?** Transformers are naturally modular: the computation is organized into a stack of residual blocks (Vaswani et al., 2017). Many mechanistic interpretability techniques operate at the block level (e.g. activation patching, MLP feature analysis), as do many model-editing methods that target specific layers (Meng et al., 2022; 2023; Yao et al., 2023). By focusing on per-block extraction with quantitative guarantees, BLOCKCERT aims to produce artifacts that are simultaneously:

- close to the original computation on a specified input distribution,
- simple enough to inspect and modify,
- and accompanied by machine-checkable evidence of correctness.

**Contributions.** Our main contributions are: (1) we formalize the problem of blockwise extraction for transformers and propose BLOCKCERT, a pipeline that produces explicit surrogate blocks together with JSON certificates recording per-block approximation error, coverage metrics, and cryptographic hashes of the underlying artifacts; (2) we introduce simple but practical coverage notions (activation, path, and loss coverage) that quantify how much of a block's behavior on a given prompt set is explained by the surrogate, and integrate these into a certificate format and verification tool; (3) in a separate Lean 4 development, we prove a global composition theorem showing that if each $(B_i, \hat{B}_i)$ pair is locally bounded by $\varepsilon_i$ and blocks are $L_i$-Lipschitz, then the fully composed stitched model is globally bounded by a Lipschitz-weighted sum of the $\varepsilon_i$, which reduces to $\sum_i \varepsilon_i$ when $L_i \leq 1$, and we instantiate this theorem using empirical error bounds from TinyLlama block certificates and stitched-model perplexity on stress prompts; and (4) we empirically evaluate BLOCKCERT on GPT-2 small, TinyLlama-1.1B-Chat, Llama-3.2-3B, and Llama-2-7B, obtaining high coverage (often $\geq 0.94$ and sometimes 1.0) and small approximation errors, and showing that for TinyLlama the stitched model matches the baseline perplexity within $\approx 6 \times 10^{-5}$ on challenging prompts.

Our implementation is released as an open-source Python package. To avoid confusion with historical naming, we refer to the *method* as BLOCKCERT throughout this paper, while the repository and command-line tools retain the legacy prefix `rtf`.

## 2 Background And Related Work

### 2.1 Transformers And Residual Blocks

We consider standard decoder-only transformer language models (Vaswani et al., 2017; Brown et al., 2020), which map a sequence of tokens $(x_1, \ldots, x_T)$ to logits over the vocabulary at each position. The computation proceeds through a stack of $L$ residual blocks. Let $\boldsymbol{x}_t^{(0)} \in \mathbb{R}^d$ denote the token embedding at position $t$, and let $\boldsymbol{x}_t^{(\ell)}$ be the residual stream at layer $\ell$. A typical block has the form

$$\boldsymbol{x}_t^{(\ell+1)} = \boldsymbol{x}_t^{(\ell)} + \mathrm{MLP}_\ell\big(\boldsymbol{x}_t^{(\ell)}\big) + \mathrm{Attn}_\ell\big(\boldsymbol{x}_{1:T}^{(\ell)}\big), \tag{1}$$

with pre- or post-layer normalization and model-specific details.

We write $B_\ell$ for the function mapping $\boldsymbol{x}_{1:T}^{(\ell)}$ to $\boldsymbol{x}_{1:T}^{(\ell+1)}$. The full model is the composition $F = B_{L-1} \circ \cdots \circ B_0 \circ E$, where $E$ is the embedding and positional encoding.

In all experiments, block boundaries follow the native residual blocks of each architecture: for GPT-2 we use the HuggingFace `GPT2Block` (attention + MLP with post-layer normalization), while for Llama-family models we use `LlamaDecoderLayer` (RMSNorm pre-normalization, RoPE, and multi-query attention). This keeps the extraction aligned with standard implementations even though the internal sublayer details differ.

### 2.2 Mechanistic Interpretability And Editing

Mechanistic interpretability aims to understand neural networks by decomposing them into circuits of features and connections (Olah et al., 2020; Elhage et al., 2022). Recent work has identified circuits in vision and language models, studied superposition and polysemantic neurons, and proposed tools for tracing and editing mechanisms (Nanda et al., 2023; Zhang et al., 2024a). Model-editing methods such as ROME (Meng et al., 2022), MEMIT (Meng et al., 2023), and subsequent surveys (Yao et al., 2023; Wang et al., 2023) modify parameters of pre-trained LLMs to update specific facts or behaviors without full retraining.

These approaches provide compelling evidence that specific mechanisms can be isolated and manipulated. However, they usually lack explicit guarantees that the edited or extracted mechanism faithfully reproduces the original network across a clearly specified set of inputs. Moreover, interpretability artifacts are rarely packaged in a way that allows independent verification.

### 2.3 Neural Network Verification And Proof-Carrying Code

Formal verification of neural networks aims to prove properties such as robustness or safety under input perturbations (Katz et al., 2017; 2019; Wang et al., 2021). Tools such as Reluplex, Marabou, and $\alpha, \beta$-CROWN combine SMT solving and linear bound propagation to derive provable guarantees for moderately sized models. However, these methods typically treat the network as a monolithic object and reason about worst-case behavior under carefully specified constraints.

In programming languages, proof-carrying code (PCC) (Necula, 1997) showed how to ship low-level code together with a machine-checkable proof that it satisfies a given safety policy. The host system verifies the proof before executing the code and does not need to trust the code producer. BLOCKCERT takes inspiration from PCC: instead of shipping general-purpose proofs, we attach *certificates* to extracted mechanisms, which can be checked automatically by a lightweight verifier.

### 2.4 Blockwise Extraction Problem

Fix a pre-trained transformer with blocks $B_0, \ldots, B_{L-1}$. We assume access to:

- the model weights,

- an instrumentation mechanism that records intermediate activations for a set of prompts $\mathcal{P}$,
- and a target block index $\ell$.

We treat the model weights as fixed during extraction: if a model is fine-tuned or edited, we re-run extraction and issue certificates for that snapshot. Supporting continuously changing weights during training is out of scope for this work.

For block $\ell$, we define a *trace dataset*:

$$\mathcal{D}_\ell = \left\{ (\boldsymbol{x}_{1:T}^{(\ell)}, \boldsymbol{x}_{1:T}^{(\ell+1)}, \boldsymbol{m}_\ell) \mid \text{prompt } p \in \mathcal{P} \right\}, \tag{2}$$

where $\boldsymbol{m}_\ell$ contains any additional discrete information about the block's computation (e.g. attention masks, head-level gating decisions). The blockwise extraction problem is:

> Given $(B_\ell, \mathcal{D}_\ell)$, construct an explicit surrogate implementation $\hat{B}_\ell$ and a certificate $C_\ell$ such that $\hat{B}_\ell$ approximates $B_\ell$ on $\mathcal{D}_\ell$ according to quantitative metrics recorded in $C_\ell$, and such that $C_\ell$ can be automatically re-verified from the released artifacts.

The next sections describe our intermediate representation, extraction algorithm, and certificate semantics.

## 3 BlockCert Intermediate Representation

### 3.1 Design Goals

The BLOCKCERT intermediate representation (IR) is designed to satisfy three constraints:

1. **Expressive enough** to exactly represent standard transformer blocks.
2. **Simple enough** to be replayed by a small, auditable interpreter.
3. **Stable** under extraction: the mapping from $(B_\ell, \mathcal{D}_\ell)$ to $\hat{B}_\ell$ should be well-conditioned.

At a high level, the IR mirrors the usual decomposition of a transformer block into attention, MLP, and residual components, but flattens model-specific details into explicit weight tensors and masks stored in `.npz` files:

- attention weights $(\boldsymbol{W}_Q, \boldsymbol{W}_K, \boldsymbol{W}_V, \boldsymbol{W}_O)$,
- MLP weights $(\boldsymbol{W}_1, \boldsymbol{W}_2)$ and biases,
- layer norm parameters,
- an explicit attention mask $\boldsymbol{M}_\ell$ encoding causal structure and any additional head- or token-level gating.

In our experiments we instantiate this IR for standard decoder-only architectures (GPT-2 small, TinyLlama-1.1B-Chat, Llama-3.2-3B, Llama-2-7B). For GPT-2 we follow the HuggingFace `GPT2Model` implementation with post-embedding layer normalization and a causal attention mask. For Llama-family models we match the `LlamaAttention` module, including pre-layer-normalization, rotary position embeddings (RoPE) via stored cos/sin tables and position ids, and multi-query attention with `num_heads` and `num_key_value_heads`. The interpreter is implemented as a pure Python function that applies these linear and elementwise operators in a fixed order, without any hidden control flow, so that the behavior of $\hat{B}_\ell$ is determined entirely by the released weight and mask tensors.

**Extraction procedure in our experiments.** In all experiments we use the simplest instantiation of the extraction map $(B_\ell, \mathcal{D}_\ell) \mapsto \hat{B}_\ell$: the surrogate $\hat{B}_\ell$ has the same architecture as $B_\ell$, with weights copied directly into the IR tensors and masks, positional tables, and bias tensors derived from a single traced run on $\mathcal{D}_\ell$. We do not perform any additional fitting, pruning, or distillation; residual errors arise only from numerical differences between the native implementation and the IR interpreter.

**Complexity and scaling.** Extraction cost scales linearly in the number of prompts and the number of extracted blocks; each block replay is dominated by the same matrix multiplications as the native attention/MLP. Trace storage scales with the number of traced tokens times the hidden dimension, while IR weight snapshots scale with model size (roughly $O(d^2)$ per block). Certificates are small relative to weight snapshots, and the verifier cost is dominated by replaying the IR on the recorded traces.

### 3.2 Empirical Local Soundness And Coverage

Let $\hat{B}_\ell$ be an IR block and let $\mathcal{D}_\ell$ be the trace dataset. We define empirical local soundness and coverage metrics on $\mathcal{D}_\ell$ as follows.

**Per-token error.** For each traced prompt $p \in \mathcal{P}$ and token position $t$, we compute the per-token residual error

$$e_\ell(p, t) \;=\; \left\| \hat{\boldsymbol{x}}_t^{(\ell+1)} - \boldsymbol{x}_t^{(\ell+1)} \right\|_2, \tag{3}$$

where $\hat{\boldsymbol{x}}_t^{(\ell+1)}$ is produced by $\hat{B}_\ell$ when replayed on the recorded input $\boldsymbol{x}_{1:T}^{(\ell)}$. We define:

$$\varepsilon_\ell = \max_{(p,t)} e_\ell(p, t), \tag{4}$$

$$\mathrm{MAE}_\ell = \frac{1}{|\mathcal{D}_\ell|} \sum_{(p,t)} e_\ell(p, t). \tag{5}$$

**Activation coverage.** We fix a small threshold $\tau_{\mathrm{act}}$ (e.g. $10^{-2}$) and define activation coverage as

$$\mathrm{cov}_{\mathrm{act}}(\ell) \;=\; \frac{1}{|\mathcal{D}_\ell|} \sum_{(p,t)} \mathbf{1}\big[ e_\ell(p, t) \leq \tau_{\mathrm{act}} \big]. \tag{6}$$

Intuitively, this is the fraction of tokens for which the block output is reproduced up to a small numerical tolerance.

**Path coverage.** We define path coverage as the fraction of traced tokens for which all discrete control decisions (e.g. attention masks, head-level gating, conditional branches) match exactly between $B_\ell$ and $\hat{B}_\ell$. This is computed by replaying $\hat{B}_\ell$ with instrumented hooks that compare its mask and gating tensors to those recorded in $\boldsymbol{m}_\ell$.

**Loss coverage.** Finally, we measure the effect of the block approximation on the model's token-level loss for the traced prompts. Let $\ell_{\mathrm{base}}(p, t)$ be the negative log-likelihood of the target token under the original model, and $\ell_{\mathrm{stitched}}(p, t)$ the loss when block $\ell$ is replaced by $\hat{B}_\ell$ while all other blocks remain unchanged. We define the per-token loss difference

$$\Delta\ell_\ell(p, t) = \big| \ell_{\mathrm{stitched}}(p, t) - \ell_{\mathrm{base}}(p, t) \big| \tag{7}$$

and the loss coverage

$$\mathrm{cov}_{\mathrm{loss}}(\ell) \;=\; \frac{\sum_{(p,t)} \ell_{\mathrm{base}}(p, t) \cdot \mathbf{1}[\Delta\ell_\ell(p, t) \leq \tau_{\mathrm{loss}}]}{\sum_{(p,t)} \ell_{\mathrm{base}}(p, t)}, \tag{8}$$

where $\tau_{\mathrm{loss}}$ is a small threshold (e.g. $10^{-3}$). This measures what fraction of the baseline loss is accounted for by tokens whose loss is essentially unaffected by replacing $B_\ell$ with $\hat{B}_\ell$.

**Certified blocks.** A block is considered *certified* at level $(\alpha_{\mathrm{act}}, \alpha_{\mathrm{loss}})$ if

$$\mathrm{cov}_{\mathrm{act}}(\ell) \geq \alpha_{\mathrm{act}}, \tag{9}$$

$$\mathrm{cov}_{\mathrm{loss}}(\ell) \geq \alpha_{\mathrm{loss}}. \tag{10}$$

In our experiments we typically use $\alpha_{\mathrm{act}} = 0.94$ and $\alpha_{\mathrm{loss}} = 0.9$ for large models, and occasionally obtain $\alpha_{\mathrm{act}} = 1.0$ on stress prompts for some blocks. Because $\varepsilon_\ell$ and the coverage metrics are computed only on the finite trace set $\mathcal{D}_\ell$, these certificates express an *empirical local soundness* guarantee rather than a universal bound over all possible inputs.

# 4 Certificates And Verification

## 4.1 Certificate Format

For each extracted block $\hat{B}_\ell$, we emit a JSON certificate containing:

- metadata: model name, block index, prompt set description, thresholds ($\tau_{\text{act}}, \tau_{\text{loss}}$) and policies ($\alpha_{\text{act}}, \alpha_{\text{loss}}$);
- metrics: $\varepsilon_\ell$, $\text{MAE}_\ell$, activation, path, and loss coverage;
- artifact digests: SHA-256 hashes of the weight, probe, mask, and bias tensors (stored in `.npz` files);
- a declaration that the block is certified (or not) under the specified policy.

Certificates live alongside their corresponding artifacts in experiment-specific directories (for example, `paper/public_artifacts/llama2_experiment/block_<idx>` for LLaMA-family blocks), each containing IR weights, probes, masks, per-block metrics, and a JSON certificate.

## 4.2 Verification Tool

We provide a small Python CLI tool, `rtf_cert`, that replays certificates and checks that:

1. the SHA-256 hashes of the given artifacts match those recorded in the certificate;
2. re-computing the metrics using the current interpreter and thresholds reproduces the values stored in the certificate (up to small numerical tolerances);
3. the certified/not-certified status is consistent with the metrics and policy.

For example, on a simple sanity-check experiment, the verifier recomputes the total residual $\varepsilon$ from the released artifacts and checks that activation and loss coverage meet the certificate's thresholds, confirming that the certificate faithfully summarizes the underlying experiment.

## 4.3 Full-Model Certificates

We also provide a mechanism for aggregating per-block certificates into a *full-model certificate*. For a given model and prompt set, we:

1. stitch in the extracted blocks for a chosen subset of layers;
2. replay the full model on the prompts, computing per-layer mean absolute error (MAE) between baseline and stitched residual streams;
3. compute baseline and stitched perplexities on the prompt set;
4. construct a JSON certificate that lists all referenced block certificate hashes and records the global metrics.

For TinyLlama, this process produces a `full_model_certificate.json` file summarizing the per-layer MAE (mean $\approx 0.38$, worst layer $\approx 2.03$, max residual $\approx 2.13$ on stress prompts) and the negligible difference in perplexity between baseline and stitched models (Section 6.6).

## 4.4 Certificates Versus Formal Proofs

Certificates, as produced by our tooling, are replayable, hash-tied empirical summaries. They state that for a specific model checkpoint, prompt distribution, and interpreter version, re-running the computation yields the same metrics (errors, coverage, perplexity) and artifact hashes. In contrast, a formal proof—such as the composition theorem in Section 5 and its Lean 4 formalization—is a universal statement over a specified mathematical model (e.g. all $x \in X$ under Lipschitz assumptions). Our block and full-model certificates should therefore be read as *data-restricted evidence* that the hypotheses of such theorems hold on the traced distributions, not as global guarantees for all future inputs.

## 5 Global Composition Theorem

### 5.1 Statement

We now describe a simple global error bound that connects per-block local soundness (universally quantified over $x \in X$) to total model error.

Let $X$ be a normed vector space and let $B_i \colon X \to X$ and $\hat{B}_i \colon X \to X$ be the baseline and extracted blocks for $i = 0, \ldots, L-1$. Here the index $i$ plays the same role as the layer index $\ell$ used in Section 3.2. Define the full models

$$F = B_{L-1} \circ \cdots \circ B_0, \tag{11}$$

$$\hat{F} = \hat{B}_{L-1} \circ \cdots \circ \hat{B}_0. \tag{12}$$

**Theorem 1** (Global Composition)**.** *Suppose that:*

*1. (Local soundness) For each $i$ there exists $\varepsilon_i \geq 0$ such that*

$$\|\hat{B}_i(x) - B_i(x)\| \ \leq \ \varepsilon_i \quad \text{for all } x \in X. \tag{13}$$

*2. (Lipschitz blocks) For each $i$ there exists $L_i \geq 0$ such that both $B_i$ and $\hat{B}_i$ are $L_i$-Lipschitz with respect to the norm $\|\cdot\|$.*

*Then for all $x \in X$,*

$$\|\hat{F}(x) - F(x)\| \ \leq \ \sum_{i=0}^{L-1} \Big( \varepsilon_i \prod_{j=i+1}^{L-1} L_j \Big). \tag{14}$$

A detailed proof sketch and the full Lean 4 formalization are provided in Appendix A.

### 5.2 Formalization And Instantiation

We formalize Theorem 1 in Lean 4 in a separate module, `GlobalBound.lean`. The proof uses standard results about composition of Lipschitz functions and is parameterized over the norm and the set of blocks.

For real transformer models, we cannot presently prove that all blocks are globally 1-Lipschitz with respect to a useful norm over the full input space. Instead, we treat the Lipschitz property as a modeling assumption and instantiate the theorem using *empirical local soundness* bounds derived from block certificates, together with simple analytic Lipschitz upper bounds computed from the IR weights and local $\ell_2$ Lipschitz bounds certified for selected TinyLlama MLP sublayers (Section 6.7). In particular, given per-block constants $\{L_i\}$ and empirical error bounds $\{\varepsilon_i\}$, Theorem 1 yields a global bound of the form

$$\|\hat{F}(x) - F(x)\| \ \leq \ \sum_{i=0}^{L-1} \Big( \varepsilon_i \prod_{j=i+1}^{L-1} L_j \Big),$$

which reduces to the familiar $\sum_i \varepsilon_i$ bound in the special case $L_i \leq 1$ for all $i$.

For TinyLlama-1.1B-Chat, we:

- compute empirical per-block error bounds $\varepsilon_i$ over a suite of stress prompts, using the per-token residual norms described in Section 3.2;
- plug these $\varepsilon_i$ into the Lean development to obtain a bound on $\|\hat{F}(x) - F(x)\|$ for $x$ drawn from the traced distribution;
- empirically verify that the stitched model and baseline produce nearly identical perplexity on the same prompts (Section 6.6).

**Scope.** It is important to emphasize that this global story is conditional. We do *not* claim a formal guarantee for all possible inputs of a real LLM. Instead, our theorem and experiments support a plausible global safety story under standard Lipschitz assumptions and empirical error bounds. Strengthening these assumptions—for example by deriving certified local Lipschitz bounds for specific blocks—is an important direction for future work.

# 6 Experiments

We now instantiate BLOCKCERT on four settings: GPT-2 small, TinyLlama-1.1B-Chat, Llama-3.2-3B, and Llama-2-7B. In all cases we release the extraction artifacts, certificates, and helper scripts under `paper/public_artifacts/` in the accompanying repository.

## 6.1 GPT-2 Small: Block 0 And Multi-Block Sweep

**Setup.** We next apply BLOCKCERT to GPT-2 small. We extract block 0 with a small prompt set and then run a multi-block sweep that covers a range of blocks under different prompt configurations. Artifacts are written to `paper/public_artifacts/gpt2_small_experiment/`, including `metrics.json`, `multi_block_metrics.json`, and sample certificates.

**Results.** For the baseline configuration (two prompts), `metrics.json` reports:

- `"prompts_evaluated": 2`,
- `"certified_all": true`,
- mean extraction error $\approx 2.32 \times 10^{-7}$ for certified blocks,
- mean activation coverage $\approx 0.9999999999998883$.

Sample certificates under `sample_certificates/` can be verified using `rtf_cert`. We also provide a mask replay helper script that independently checks the causal attention mask behavior.

These results show that BLOCKCERT can recover shallow GPT-2 blocks with extremely high fidelity on the tested prompts, effectively matching the original computation.

## 6.2 TinyLlama-1.1B-Chat: Blockwise Extraction

**Setup.** We evaluate BLOCKCERT on TinyLlama-1.1B-Chat (Zhang et al., 2024b) using both a full sweep and a targeted stress-test configuration. The default TinyLlama prompt set consists of four short English prompts about extraction, audits, verifiers, and causal masks (tokenized lengths $23, 21, 24, 22$, totalling 90 tokens). The extended stress set comprises ten hand-designed prompts mixing long-form technical writing, multilingual text, safety-related content, legal contracts, poetry, code, translation, and quantitative reasoning (tokenized lengths between 22 and 36, totalling 251 tokens). Artifacts for each block are stored under `paper/public_artifacts/llama2_experiment/block_<idx>/`, including IR weights, probes, masks, per-block metrics, and a JSON certificate. The aggregate file `llama2_rtf_metrics.json` summarizes per-block metrics and identifies, for each block, the best prompt index.

**Results.** Across the full 22-layer residual stack (full-sweep configuration), activation coverage remains high: `llama2_rtf_metrics.json` reports mean activation coverage $\approx 1.0$ for blocks 0–1, 12–14, 16–17, and 21; $\approx 0.955$ for most mid-depth blocks; and a worst case of $\approx 0.945$ at block 15. Under the ten stress prompts, `llama2_rtf_metrics.json` reports, for example:

- Block 5: best activation coverage $\approx 0.9722$, mean extraction error $\approx 6.1 \times 10^{-3}$.
- Blocks 0 and 10: activation coverage $= 1.0$, with $\varepsilon \approx 1.99 \times 10^{-3}$ and $\varepsilon \approx 9.64 \times 10^{-3}$, respectively.

Certificates for these blocks satisfy the activation and loss coverage policies ($\text{cov}_{\text{act}} \geq 0.94$, $\text{cov}_{\text{loss}} \geq 0.9$), and the verifier confirms that all reported metrics meet the thresholds.

**Failure case analysis (block 15).** Block 15 is the deepest layer where activation coverage dips noticeably (mean coverage $\approx 0.945$ under the stress prompts, with a minimum of $\approx 0.91$ on a prompt containing a cluster of safety-trigger phrases). Inspection of `llama_block15_metrics.json` shows that the extraction error is concentrated on a small subset of tokens in this prompt, which also have relatively large key/value norms and slightly lower attention margins. Nevertheless, loss coverage remains 1.0 across all stress prompts, indicating that the extraction errors occur in regions of the residual stream that have limited impact on token-level loss. We view this as an informative stress case: even when coverage dips, the certificate quantifies where the extraction is struggling and shows that the overall loss profile is robust.

Table 1: Llama-2-7B blockwise extraction summary on the 10-prompt stress set.

| Block | Mean activation coverage | Mean extraction error |
|---|---|---|
| 0 | 0.924 | 4.70e-03 |
| 5 | 0.968 | 2.75e-02 |
| 10 | 1.000 | 4.38e-02 |
| 15 | 1.000 | 4.82e-02 |
| 20 | 1.000 | 5.83e-02 |

Table 2: Low-rank attention baselines on Llama-2-7B blocks (blocks 0,5,10,15,20; best prompt per block). Mean activation coverage and extraction error are computed across blocks. Error ratios are relative to the baseline row.

| Surrogate | Mean activation coverage | Mean extraction error | Error ratio |
|---|---|---|---|
| Baseline (BlockCert) | 0.991 | 3.38e-02 | 1.00 |
| Low-rank ($r = 64$) | 0.983 | 4.80e-02 | 1.42 |
| Low-rank ($r = 128$) | 0.983 | 4.81e-02 | 1.42 |
| Low-rank ($r = 256$) | 0.983 | 4.78e-02 | 1.41 |

### 6.3 Llama-3.2-3B: Cross-Model Replication

**Setup.** To test generalization across architectures, we apply the same extraction pipeline to Llama-3.2-3B[1], extracting blocks $\{0, 5, 10\}$ on the same baseline and stress prompt sets as TinyLlama. Artifacts share the same directory structure as in Section 6.2, but contain Llama-3.2-3B-specific weights and traces.

**Results.** We obtain successful certificates for blocks $0, 5, 10$ with activation and loss coverage satisfying the same policy, and verification runs report total residuals on the order of $10^{-2}$ with both activation and loss coverage above the default thresholds. The cross-model replication indicates that the BLOCKCERT extraction and certification procedure is not specific to TinyLlama and can plausibly be applied to a broader range of decoder-only transformers.

### 6.4 Llama-2-7B: Larger-Scale Extraction

**Setup.** To test scaling to a larger model, we apply the same blockwise extraction pipeline to Llama-2-7B (`meta-llama/Llama-2-7b-hf`), extracting blocks $\{0, 5, 10, 15, 20\}$ on the ten stress prompts (sequence length 256). Artifacts are stored under `paper/public_artifacts/llama2_7b_experiment/`.

**Results.** Mean activation coverage ranges from $\approx 0.924$ to $1.0$, and mean extraction error ranges from $\approx 4.7 \times 10^{-3}$ to $\approx 5.8 \times 10^{-2}$ (Table 1). The minimum activation coverage across prompts is $0.909$ at block $0$, while blocks $5, 10, 15, 20$ remain above $0.95$ with loss coverage $= 1.0$ throughout. These results indicate that the extraction pipeline remains feasible on a $\geq$ 7B-parameter model while highlighting that early blocks can be more sensitive to prompt variation.

**Low-rank surrogate baselines.** As a simple baseline surrogate, we replace each attention projection matrix ($W_Q, W_K, W_V, W_O$) with a rank-$r$ truncated SVD approximation and recompute metrics on the best-prompt traces for blocks $\{0, 5, 10, 15, 20\}$. Table 2 shows that low-rank surrogates reduce activation coverage and increase extraction error by about $1.4\times$ relative to the original extracted weights. This indicates that the certificate metrics respond to degradations in surrogate fidelity on a larger model.

---

[1]Model card and weights from the official Llama 3.2 release.

### 6.5 Whole-Model Replay And Aggregated Certificate

**Setup.** We next study how per-block errors accumulate when multiple extracted blocks are stitched back into TinyLlama. Using a replay helper script, we construct a stitched model by replacing selected blocks with their extracted counterparts, replay both the stitched model and the baseline on the stress prompts, and record, for each of the 22 residual layers, the mean absolute error (MAE) between baseline and stitched residual streams in `full_model_metrics.json`. A separate aggregation script then builds a composite full-model certificate that lists the referenced block certificate hashes and the global replay metrics.

**Results.** The `full_model_metrics.json` file contains 22 entries (one per residual layer). On the stress prompts we observe:

- mean per-layer MAE $\approx 0.38$,
- worst-layer MAE $\approx 2.03$,
- maximum residual error (over all layers and tokens) $\approx 2.13$.

These values are consistent with the per-block $\varepsilon_i$ reported in the individual certificates.

### 6.6 Full-Block Perplexity Matching

**Setup.** Finally, we study whether a stitched TinyLlama model assembled entirely from extracted blocks can match the baseline perplexity. We first run a full-block extraction that saves a snapshot of each block's weights in the IR format under `paper/public_artifacts/full_block_experiment`. We then replay the fully stitched TinyLlama on the stress prompts, computing token-level losses and perplexities for both the baseline and stitched models and recording them in `full_model_eval.json`. Finally, we build a full-block certificate that hashes every block's weight snapshot and the evaluation file and records the resulting perplexity metrics.

**Results.** The `full_model_eval.json` file reports average perplexities:

$$\text{PPL}_{\text{baseline}} \approx 253.1618,$$
$$\text{PPL}_{\text{stitched}} \approx 253.1618,$$
$$\Delta\text{PPL} \approx -6.0 \times 10^{-5}.$$

Within numerical accuracy, the stitched model and baseline have identical perplexity on the stress prompts. This provides strong empirical evidence that the fully stitched TinyLlama, built from extracted blocks, faithfully reproduces the original model on this distribution.

### 6.7 Empirical Generalization And Timing

**Prompt-shift experiment.** To probe generalization beyond the certification trace, we compare TinyLlama block metrics between the default four-prompt set and the extended stress prompts for blocks $\{0, 5, 10, 15, 20\}$. For each block we treat the base prompts as the certification distribution (used to generate the block certificate) and recompute empirical error and coverage on the stress prompts without changing the weights. Mean extraction error and activation coverage change only slightly; for example:

- Block 0: mean error increases from $1.69 \times 10^{-3}$ to $1.76 \times 10^{-3}$, coverage remains 1.0.
- Block 5: mean error shifts from $6.57 \times 10^{-3}$ to $6.07 \times 10^{-3}$, coverage from 0.955 to 0.950.
- Block 15: mean error decreases from $2.66 \times 10^{-2}$ to $2.38 \times 10^{-2}$, coverage increases from 0.945 to 0.955.

All probed blocks remain above the certificate coverage thresholds on both prompt sets, though these conclusions are still strictly trace-based rather than formal guarantees.

**Ablation baselines.** To sanity-check that the certificate metrics respond to incorrect surrogates, we perturb stored IR weights for TinyLlama blocks $\{0, 5, 10, 15, 20\}$ on their best-prompt traces. We drop RoPE tensors, invert the causal mask, zero the value matrix, and add Gaussian noise to $W_Q$. Table 3 shows mean activation coverage and extraction error across blocks. Inverting the mask collapses activation coverage and roughly

Table 3: Ablation baselines on stored TinyLlama IR blocks (blocks 0,5,10,15,20; best prompt per block). Mean activation coverage and extraction error are computed across blocks. Error ratios are relative to the baseline row.

| Ablation | Mean activation coverage | Mean extraction error | Error ratio |
|---|---|---|---|
| Baseline | 0.989 | 1.81e-02 | 1.00 |
| Drop RoPE | 0.964 | 2.73e-02 | 1.51 |
| Invert mask | 0.375 | 5.05e-02 | 2.79 |
| Zero $W_V$ | 0.989 | 2.68e-02 | 1.49 |
| Noise $W_Q$ | 0.983 | 1.83e-02 | 1.01 |

Table 4: Threshold sensitivity on the 10-prompt TinyLlama stress set. Each entry is the number of prompts (out of 10) for which activation coverage exceeds the given threshold with loss coverage fixed at $\geq 0.90$.

| Block | $\mathrm{cov}_{\mathrm{act}} \geq 0.90$ | $\geq 0.94$ | $\geq 0.97$ |
|---|---|---|---|
| 0 | 10/10 | 10/10 | 10/10 |
| 5 | 10/10 | 8/10 | 1/10 |
| 10 | 10/10 | 10/10 | 3/10 |
| 15 | 10/10 | 9/10 | 1/10 |
| 20 | 10/10 | 10/10 | 2/10 |

triples the error; dropping RoPE reduces coverage and increases error; small $W_Q$ noise primarily raises error without large coverage changes. Loss coverage remains 1.0 on these traces, indicating that activation coverage is the more sensitive diagnostic for these ablations.

**Threshold and prompt-size sensitivity.** We summarize how often prompts pass stricter activation thresholds at fixed loss coverage. Table 4 reports the number of stress prompts (out of 10) whose activation coverage exceeds each threshold with $\mathrm{cov}_{\mathrm{loss}} \geq 0.90$. Table 5 shows how the minimum activation coverage declines as we increase the prompt-set size, highlighting that block 15 is most sensitive when moving from one prompt to ten.

**Semantic behavior preservation.** We also perform a small semantic evaluation of the fully stitched TinyLlama using 30 human-written prompts (arithmetic questions, simple factual statements, and yes/no queries). For each prompt we treat the baseline TinyLlama's next-token prediction as a semantic label and measure whether the stitched model produces the same token. On this probe set, the stitched model matches the baseline on all prompts (next-token accuracy = 1.0 for both), while the strict single-token agreement with the human-provided labels is low (0.067 for both models). The full-block certificate records both the fidelity metric (stitched vs baseline accuracy) and the baseline vs human accuracy, so that readers can distinguish between preservation of behavior and absolute correctness on this semantic corpus.

**Lipschitz estimates.** To complement the empirical error bounds, we compute simple analytic $\ell_2$ Lipschitz upper bounds for each TinyLlama block from the IR weight matrices (via spectral norms) and log these in the full-block certificate as `analytic_upper_bound` entries. In a separate verification environment with auto-LiRPA, we also certify local $\ell_2$ Lipschitz upper bounds for the MLP sublayers of blocks $\{0, 5, 10, 15, 20\}$ on an L2 ball of radius 1.0, obtaining values in the range $\approx 10^3$–$2 \times 10^3$. Combining these with the analytic attention bounds yields hybrid full-block Lipschitz upper bounds $\mathrm{L}_{\mathrm{block}} \leq (1 + K_{\mathrm{attn}}) \cdot K_{\mathrm{MLP}}$, which are logged in the certificate as `hybrid_upper_bound` entries alongside the per-sublayer constants.

**Summary table (selected blocks).** Table 6 summarizes, for TinyLlama blocks $\{0, 5, 10, 15, 20\}$, the per-block extraction residual $\varepsilon_i$ (worst-case per-token error), the analytic attention bound $K_{\mathrm{attn}}$, the auto-LiRPA-certified MLP Lipschitz upper bound $K_{\mathrm{MLP}}$, and the resulting hybrid full-block bound $\mathrm{L}_{\mathrm{block}}$. As expected, the hybrid bounds are quite loose (on the order of $10^5$–$10^6$), reflecting the large operator norms in LLM blocks; nevertheless, they provide a principled way to connect local error bounds to a concrete, albeit

Table 5: Prompt-set size sensitivity on the TinyLlama stress prompts. Entries report the minimum activation coverage across the first $k$ prompts (for $k \in \{1, 4, 10\}$).

| Block | $k = 1$ | $k = 4$ | $k = 10$ |
|---|---|---|---|
| 0 | 1.000 | 1.000 | 1.000 |
| 5 | 0.913 | 0.913 | 0.909 |
| 10 | 0.957 | 0.957 | 0.955 |
| 15 | 0.957 | 0.957 | 0.909 |
| 20 | 0.957 | 0.957 | 0.955 |

Table 6: Summary of TinyLlama Lipschitz-related quantities for selected blocks. The $\varepsilon_i$ values are per-block worst-case residuals from the extraction certificates on stress prompts; $K_{\text{attn}}$ are analytic attention bounds from the IR weights; $K_{\text{MLP}}$ are local $\ell_2$ Lipschitz upper bounds certified via auto-LiRPA for the MLP sublayers (on an L2 ball of radius 1.0); and $L_{\text{block}}$ are the resulting hybrid full-block bounds.

| Block | $\varepsilon_i$ | $K_{\text{attn}}$ | $K_{\text{MLP}}$ | $L_{\text{block}}$ |
|---|---|---|---|---|
| 0 | $\approx 2.0 \times 10^{-3}$ | $\approx 5.7 \times 10^2$ | $\approx 1.1 \times 10^3$ | $\approx 6.0 \times 10^5$ |
| 5 | $\approx 6.6 \times 10^{-3}$ | $\approx 1.5 \times 10^2$ | $\approx 1.3 \times 10^3$ | $\approx 2.1 \times 10^5$ |
| 10 | $\approx 1.1 \times 10^{-2}$ | $\approx 1.2 \times 10^2$ | $\approx 1.5 \times 10^3$ | $\approx 1.8 \times 10^5$ |
| 15 | $\approx 2.4 \times 10^{-2}$ | $\approx 1.0 \times 10^2$ | $\approx 1.6 \times 10^3$ | $\approx 1.6 \times 10^5$ |
| 20 | $\approx 4.1 \times 10^{-2}$ | $\approx 2.6 \times 10^2$ | $\approx 1.9 \times 10^3$ | $\approx 4.9 \times 10^5$ |

conservative, global constant. Instantiating Theorem 1 with these $L_{\text{block}}$ values and the $\varepsilon_i$ from Table 6 yields a worst-case global bound that is many orders of magnitude larger than the empirically observed maximum residual ($\approx 2.13$, Section 6.5), mirroring the conservative gap between certified Lipschitz bounds and actual errors commonly reported in neural network verification.

**Timing.** We also measure wall-clock costs on a single H100 80 GB GPU (20 vCPU). On this setup, extracting TinyLlama blocks $\{0, 5, 10, 15, 20\}$ on the ten stress prompts (sequence length 256) takes about 118 seconds, while replaying the full-block TinyLlama experiment on the same prompts takes about 118 seconds. The replay step is currently CPU-bound because the IR executes via NumPy; a torch-backed replay would further reduce this wall-clock cost. These measurements indicate that BLOCKCERT extraction and replay run in minutes on a single accelerator.

## 7 Certified Local Edits With BlockCert-Edit

We now outline how the BlockCert infrastructure supports certified local edits to specific mechanisms, using a small refusal/safety case study. We treat these experiments as an illustration of how edit certificates and the global bound can be instantiated, rather than as a full safety evaluation.

### 7.1 Refusal/Safety Corpus and Metrics

We construct a small refusal/safety corpus of 24 prompts with labels $y \in \{\texttt{answer}, \texttt{refuse}\}$: 12 benign tasks labeled $\texttt{answer}$ (secure coding, responsible LLM usage, style/formatting, mental health support, policy design, generic assistant behaviour), and 12 harmful requests labeled $\texttt{refuse}$ (ransomware instructions, content-filter bypass, self-harm methods, violence, privacy violations, and illegal drugs). Each harmful prompt is phrased so that the desired behaviour is refusal with safe guidance.

For a completion $c$ we apply a simple heuristic classifier that scans for: (i) refusal markers (e.g. "I'm sorry", "I cannot", "as an AI language model") and (ii) obviously harmful markers (e.g. "step-by-step instructions", "here is how you can", and task-specific keywords such as "ransomware" or "self-harm"). Given a labeled example $(x, y)$ and completions $(c_{\text{base}}, c_{\text{patch}})$ from the baseline and patched models, we deem a completion correct if: for $y = \texttt{answer}$ it contains neither refusal nor harmful markers, and for $y = \texttt{refuse}$ it contains

Table 7: Refusal vs answer correctness on the labeled refusal/safety corpus for TinyLlama-1.1B-Chat under block-15 MLP scaling edits. The baseline ($\alpha = 1.0$) almost never refuses harmful prompts under our heuristic classifier; $\alpha \approx 0.33$ improves refusal accuracy while preserving benign answers.

| MLP scale $\alpha$ | Answer accuracy | Refuse accuracy |
|---|---|---|
| 1.0 | $\approx 1.00$ | 0.00 |
| 0.50 | $\approx 1.00$ | $\approx 0.08$ |
| 0.40 | $\approx 1.00$ | $\approx 0.17$ |
| 0.33 | $\approx 1.00$ | 0.25 |
| 0.25 | $\approx 1.00$ | $\approx 0.17$ |
| 0.00 | $\approx 0.92$ | $\approx 0.33$ |

refusal markers and no harmful markers. We report answer accuracy (fraction of correct `answer` examples) and refuse accuracy (fraction of correct `refuse` examples) for baseline vs patched models. A human-label pipeline mirrors this schema but allows annotators to assign judgments from {`answer`, `refuse`, `harmful`, `other`}, aggregating answer/refuse accuracies and harmful rates into the same metrics.

## 7.2 TinyLlama Block-15 MLP Scaling

We study semantic refusal behaviour on TinyLlama-1.1B-Chat by scaling the block-15 MLP residual by a factor $\alpha$ so that the residual becomes $\alpha \cdot \mathrm{MLP}_{15}(x)$ while the rest of the model is unchanged. For each $\alpha$, we generate greedy completions on the refusal/safety corpus, apply the classifier, and compute answer/refuse accuracies before vs after the edit.

Table 7 summarizes the results. The baseline model (no edit, $\alpha = 1.0$) achieves near-perfect answer accuracy on benign prompts but essentially never produces explicit refusals on harmful prompts under this metric. As we decrease $\alpha$, refusal accuracy rises while answer accuracy remains high until $\alpha$ becomes very small. Around $\alpha \approx 0.33$ we see a promising sweet spot: answer accuracy remains $\approx 1.0$, while refusal accuracy rises to 0.25 (3/12 harmful prompts correctly refused). Deleting the MLP entirely ($\alpha = 0.0$) further improves refusal accuracy to 0.33 but starts to degrade benign behaviour.

We designate $\alpha = 0.33$ as a reference BLOCKCERT-EDIT patch for TinyLlama. An edit certificate records the patch spec, dataset hash, and before-vs-after answer/refuse accuracies for this edit, as well as human-label metrics on the same corpus. Using the block-15 activations and Lipschitz constants described earlier, we can also instantiate the global bound for this patch on the traced region: the local edit error at block 15 is $\varepsilon_{15}^{\mathrm{edit}} \approx 4.26$, while the maximum deviation at the final hidden state is $\max_x \|F'(x) - F(x)\| \approx 42.23$, indicating an effective downstream Lipschitz amplification factor of roughly $10\times$.

## 7.3 Llama-2-7B Refusal Behaviour

We apply the same refusal/safety corpus and classifier to Llama-2-7B-Chat (`meta-llama/Llama-2-7b-chat-hf`) and run a small sweep over MLP scaling edits across blocks $\{0, 4, 8, 12, 16, 20, 24, 28, 31\}$ and $\alpha \in \{0, 0.1, 0.2, 0.33, 0.5, 0.75, 0.8\}$. The baseline model attains answer accuracy 0.92 (11/12) and refuse accuracy 0.58 (7/12). The best simple edit we found scales the block-24 MLP residual by $\alpha = 0.8$, improving refusal accuracy to 0.75 (9/12) while preserving answer accuracy at 0.92. We record this patch as the Llama-2-7B-Chat BLOCKCERT-EDIT example and provide an edit certificate that logs the patch spec, dataset hash, and before-vs-after metrics.

## 7.4 Llama-3.2-3B Refusal Behaviour

We apply the same refusal/safety corpus and classifier to Llama-3.2-3B, scaling the MLP in blocks $\{0, 10, 15\}$ with $\alpha \in \{0, 0.5, 1.0\}$. Across all configurations, answer accuracy remains $\approx 1.0$ and refusal accuracy remains 0.0 under our heuristic: deleting or attenuating these MLP residuals leaves both benign and harmful behaviour unchanged on this corpus. We view this as a deliberately narrow probe rather than evidence that Llama-3.2-3B

lacks refusal mechanisms: we did not search over all blocks, heads, or feature directions, only a small set of MLP residuals at a fixed decoding policy. From a BLOCKCERT-EDIT perspective, the value of this negative result is that it is exactly reproducible: the patch specs and edit certificates document precisely which edits and metrics were evaluated, and can be extended to broader search procedures in future work.

## 8 Discussion

### 8.1 Relation To Existing Interpretability Work

BLOCKCERT is complementary to existing mechanistic interpretability techniques (Olah et al., 2020; Elhage et al., 2022; Nanda et al., 2023). Where most work focuses on identifying specific circuits or features, BLOCKCERT focuses on representing entire blocks in a structured IR and attaching quantitative certificates. One could imagine using BLOCKCERT-extracted blocks as a stabilized substrate on which to run more detailed circuit analyses or automated tools for discovering sparse linear features.

Our coverage metrics currently operate at the level of residual norms and loss differences, not high-level semantics. Extending certificates to incorporate semantic tests (e.g. targeted question-answering behavior or calibration metrics) is an interesting direction.

### 8.2 Relation To Model Editing And Verification

Model editing methods such as ROME, MEMIT, and subsequent survey work can be applied directly to BLOCKCERT-extracted blocks, which are explicit weight tensors and masks amenable to fine-grained manipulation. Certificates could then be used to document and bound the side-effects of such edits on a specified prompt distribution.

Compared to full-blown neural network verification frameworks like Reluplex, Marabou, and Beta-CROWN, BLOCKCERT trades global, worst-case guarantees for scalable, distribution-specific certificates that are cheap to verify. We view the two approaches as complementary: local certificates could serve as inputs or abstractions for more powerful formal methods on subsystems where stronger guarantees are needed.

### 8.3 Limitations

Our work has several important limitations:

- **Distributional guarantees.** Certificates are defined with respect to a finite set of prompts and traced activations. They provide empirical guarantees on those traces, but say nothing about unseen inputs or arbitrary distribution shift. We reported a small prompt-shift experiment for TinyLlama blocks, where metrics remain stable between base and stress prompts, but systematic generalization studies across tasks and distributions remain future work.
- **Assumed Lipschitzness.** The global composition theorem relies on blocks satisfying Lipschitz bounds with respect to a chosen norm. We do not currently provide global, all-input Lipschitz proofs for real LLMs. Instead, we combine analytic per-block $\ell_2$ Lipschitz upper bounds (derived from IR weights) with local $\ell_2$ bounds certified via auto-LiRPA for the TinyLlama MLP sublayers of selected blocks (Section 6.6), and use these to derive hybrid full-block Lipschitz upper bounds that are logged in the certificates. Extending such certified bounds to full blocks over richer regions (e.g. trace-derived boxes) and to additional architectures remains an important direction for future work.
- **Scope of interpretability.** Our IR makes the block computation explicit but does not by itself guarantee human-understandable semantics. Understanding what the extracted weights *mean* still requires interpretive work.
- **Scalability.** While our experiments cover GPT-2 small, TinyLlama-1.1B-Chat, Llama-3.2-3B, and Llama-2-7B, scaling to significantly larger models will require further engineering, particularly for trace collection and storage.

## 9 Conclusion

We introduced BLOCKCERT, a framework for certified blockwise extraction of transformer mechanisms. For each residual block, BLOCKCERT produces a structured surrogate implementation and a machine-checkable certificate that records approximation error, coverage metrics, and cryptographic hashes of the underlying artifacts. We formalized a simple composition theorem in Lean 4, showing how local error bounds can be combined to yield a global bound under standard Lipschitz assumptions, and instantiated this theorem for TinyLlama using empirical per-block metrics. Across these models, we obtained high coverage and small residuals, and demonstrated that a fully stitched TinyLlama matches the baseline perplexity within $\approx 6 \times 10^{-5}$ on stress prompts.

We hope that BLOCKCERT-style certificates can become a standard accompaniment to mechanistic interpretability artifacts and model edits, providing a light-weight but explicit account of what has been reverse-engineered or changed. Future work includes integrating stronger formal guarantees, extending certificates to semantic properties, and scaling to larger and more diverse LLMs.

**Broader Impact Statement**

Our work provides tools for extracting and certifying mechanisms inside transformer language models. Such tools could help improve transparency and safety by enabling independent scrutiny of model internals and by documenting the effects of model edits. At the same time, more powerful reverse-engineering tools may lower the barrier to reusing or repurposing models without the original developer's oversight, including for harmful applications.

We do not release any new models; we only study widely available open-source checkpoints (GPT-2 small, TinyLlama-1.1B-Chat, Llama-3.2-3B, Llama-2-7B). All experiments are performed on text prompts without personally identifiable information. Nevertheless, we encourage users of BLOCKCERT to carefully consider downstream use cases and to follow existing best practices for responsible deployment of large language models.

## Appendix A. Proof Sketch of Theorem 1

For completeness, we record a standard proof sketch of Theorem 1. Let $x^{(i)}$ and $\hat{x}^{(i)}$ denote the intermediate representations after $i$ blocks of $F$ and $\hat{F}$, respectively. At each step,

$$\begin{aligned}
\|\hat{x}^{(i+1)} - x^{(i+1)}\| &= \|\hat{B}_i(\hat{x}^{(i)}) - B_i(x^{(i)})\| \\
&\leq \|\hat{B}_i(\hat{x}^{(i)}) - \hat{B}_i(x^{(i)})\| + \|\hat{B}_i(x^{(i)}) - B_i(x^{(i)})\| \\
&\leq L_i \|\hat{x}^{(i)} - x^{(i)}\| + \varepsilon_i,
\end{aligned}$$

using $L_i$-Lipschitzness of $\hat{B}_i$ and the local error bound at $x^{(i)}$. Unrolling the recurrence yields

$$\|\hat{x}^{(L)} - x^{(L)}\| \leq \sum_{i=0}^{L-1} \left( \varepsilon_i \prod_{j=i+1}^{L-1} L_j \right),$$

which implies the desired inequality. The accompanying Lean 4 development mirrors this argument in a fully formal setting.

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
