# OpenReview forum: "BlockCert: Certified Blockwise Extraction of Transformer Mechanisms"
_TMLR — Rejected by TMLR_

### Review · Reviewer_WtbH · 2025-12-19

**Summary Of Contributions:**

This paper proposes BlockCert, a framework for “certified blockwise extraction” of transformer mechanisms. Given a pretrained transformer and a prompt set, the method extracts per-block surrogates in an explicit intermediate representation (IR), along with JSON “certificates” that store local approximation error, coverage metrics, and hashes of the artifacts. A simple Lipschitz-based global composition theorem is formalized in Lean 4 to relate per-block error bounds to an overall deviation bound. Empirical case studies are presented on GPT-2 small, TinyLlama-1.1B-Chat, and Llama-3.2-3B, including a stitched TinyLlama model whose perplexity closely matches the baseline on a small stress set and a toy “refusal” editing experiment.

**Audience:**

Yes

**Audience Explanation:**

The work addresses an important gap between mechanistic interpretability and formal verification, and tries to make interpretability artifacts more reproducible and auditable via hash-tied certificates.

**Claims And Evidence:**

Yes

**Claims Explanation:**

The use of Lean 4 to formalize a global composition theorem is a nice step toward connecting interpretability practice with formal methods, even if the instantiated bounds are conservative.

The implementation appears lightweight and practical, and the experiments suggest that simple IR-based replay can indeed match native implementations at the block level and, for TinyLlama, at the full-model level on a small prompt set.

**Requested Changes:**

1 Explicitly position the contribution as a tooling and standardization effort (a certificate format + pipeline), instead of a novel interpretability or verification method. This may better match the actual technical content.

2 Include at least one larger model (e.g., ≥ 7B parameters) or, if not feasible, provide a careful scalability analysis (memory, time, trace size) and clearly state limitations.

3 Evaluate BlockCert in at least one realistic application scenario, e.g. editing factual associations, safety behavior, or algorithmic tasks, and show that certificates help diagnose or bound side effects compared to baselines.


4 Expand coverage of recent mechanistic interpretability, editing, and verification work.


5 Go beyond residual norms and perplexity, and include semantic fidelity metrics (e.g., agreement on downstream labels, consistency with human annotations) in the certificates.

---

### Review · Reviewer_6KNo · 2026-01-06

**Summary Of Contributions:**

This paper proposes BlockCert, a framework for certified blockwise extraction of transformer mechanisms. Given a model and a prompt set, it records the per-block traces; emits, for each block, an explicit intermediate representation (IR) implementation together with the corresponding artefacts (weights, masks, etc.); and produces a certificate, serialised in JSON (why the emphasis on the file formats? also "activations etc. are stored in NPZ files" -- isn't this just a marginal implementation detail?), containing error and coverage metrics and cryptographic hashes (SHA-256) for third parties to verify the contents. The paper provides a Lipschitz-based "Global Composition" theorem (also formalised in Lean 4) to relate per-block deviation bounds to a global deviation bound, and proposes an extension (BlockCert-Edit) for documenting and certifying local edits. Experiments on smaller-scale models like GPT-2 small, TinyLlama-1.1B, and Llama-3.2-3B show promising activation and loss coverage on the considered prompts, and for TinyLlama, a fully "stitched" model matches baseline perplexity.
I like how the paper proposes an end-to-end workflow (IR + verifier + certificates) and a theorem connecting local and global deviation (under some reasonable assumptions). However, the "extraction" bit used in the experiments basically provides a faithful reimplementation of the original blocks, and I am not sure what kind of simplified mechanisms it is extracting. Experiments revolve around tiny models with a set of curated prompt sets, and it is not clear how that could generalise to more realistic scenarios. In experiments, the paper does not seem to provide any baseline. The "Global Composition theorem" seems a standard Lipschitz error-propagation bound for compositions -- it is not novel, but its application as a bookkeeping bridge from per-block certificates to a global bound can be.

**Audience:**

Yes

**Audience Explanation:**

It is an interesting paper at the intersection between mechanistic interpretability, reproducibility/distillation, and formal methods.

**Broader Impact Concerns:**

I do not think this paper raises any ethical concerns.

**Claims And Evidence:**

Yes

**Claims Explanation:**

The paper claims that blockwise certificates for transformer-block surrogates are feasible in practice, and that we can provide some global guarantees about their composition. This is supported by the paper, both theoretically ("Global Composition Theorem") and practically (there are no baselines but the numerical results seem good/convincing).

**Requested Changes:**

The paper's story seems to be about "mechanism extraction", but the main contribution may be more in terms of "certified blockwise reimplementation"; evaluating on non-toy models would provide significant value to the paper; likewise, evaluating the reimplemented blocks and models on real-world datasets and tasks would help understand the applicability of the proposed method. How sensitive is the method to the choice of the $\tau$ and $\alpha$ hyperparams? Can you share some details on the computational complexity/runtimes?

---

### Review · Reviewer_E9hV · 2026-01-12

**Summary Of Contributions:**

In this paper, the authors propose BlockCert, a framework for certified blockwise extraction
of transformer mechanisms, which extracts structured surrogate implementations for residual blocks together with machine-checkable certificates that bound approximation error, record coverage metrics, and hash the underlying artifacts for a pre-trained transformer and a prompt distribution. Despite the certain degree of innovation demonstrated by this paper, it still suffers from notable deficiencies in theoretical depth, merits of the proposed solution, and academic writing quality. Therefore, this paper cannot be accepted due to it does not meet the journal's requirements.

**Additional Comments:**

Some typos need to be carefully checked in this paper.

**Audience:**

No

**Audience Explanation:**

1.	The main Innovativeness of this paper just likes the combination of some techniques.
2.	The absence of a Related Work section in this paper compromises the persuasiveness of its literature grounding.

**Claims And Evidence:**

No

**Claims Explanation:**

1.	This paper looks like a experimental report rather than a academical paper.
2.	The authors should present necessary figures to illustrate the proposed framework. The figure 1 is too abstract to understand their workflow.
3.	How to divide the block for a model. Do all chosen models have the same approach to divide blocks?
4.	Why limit the model to a pre-trained? This limitation significantly influences the scalability of the proposed method.
5.	The experiments are only conducted on several specific models, which makes the result unreliable.
6.	Lacking of comparison with the state-of-the art works also make the experimental result not credible.

**Requested Changes:**

None

---

### Decision · Action_Editor_tGVH · 2026-04-08

**Recommendation:** Reject

**Additional Comments:**

This paper proposes BlockCert, a framework for certified blockwise extraction of transformer mechanisms, which extracts structured surrogate implementations for residual blocks together with machine-checkable certificates that bound approximation error, record coverage metrics, and hash the underlying artifacts for a pre-trained transformer and a prompt distribution. Certified blocks can be stitched together to provide a model-level certification, and the behaviour of the model can be edited. Reviewers liked the end-to-end workflow and the use of the Lipschitz bound as a bookkeeping bridge (6KNo) and the use of Lean to formalize this as a theorem (WtbH). The method is lauded as being lightweight and practical (WtbH). Overall, reviewers also liked the placement of this paper as addressing an important gap between mechanistic interpretability and formal verification.

However,the reviewers raised some key points: placement in the literature and comparison with the state-of-the-art; a change to an emphasis on tooling; testing on larger models; adding baselines; and a realistic application scenario. Some reviewers mentioned a lack of novelty: this is not a criterion for TMLR and therefore this critique has been disregarded.

The paper has been updated with additional results on a larger model (Llama-2-7B), baselines, and a realistic application scenario, but is still lacking in certain respects, and therefore cannot be accepted in its present form. I detail necessary changes below.

1. Related work. Reviewers E9hV and WtbH both ask for the related work section to be expanded. While the paper does have a related work section, and this has now been explicitly renamed, the discussion of related work as opposed to background is really quite short. The proposed pipeline with should be compared with similar methods in more detail, and what this pipeline offers over and above previous similar approaches should be explained. Alternatively, it can also be argued that such a pipeline does not yet exist, if this is true.

2. Related to the above point, reviewer Eh9V also asks for comparison with state of the art models. While a simple low-rank approximation baseline has been added, which is good, there is still no explicit numerical comparison with existing approaches. Explicit numerical comparison with existing approaches (or, an argument as to why this is not necessary) should be given.

3. Some details of the pipeline are missing. Reviewer E9hV asks for the diagram in Figure 1 to be made more explicit. This diagram has been updated, however details of the pipeline are still missing. Specifically, an explicit description of the verifier is missing. Section 4.2 explains that a Python tool is provided, but does not explain what it means to replay a certificate. A description of this part of the pipeline should be given.

4. Making the contributions of the paper more explicit. All reviewers agree that the emphasis should be made on tooling rather than interpretability. This has been altered in the updated manuscript. In particular, reviewer 6KNo argues that since the extraction map has the same architecture as the original model, this provides more of a pipeline than an extraction mechanism. Expanding on this point, I would like to see some more explanation of what the certification offers: if the $\hat{B}_i$ carry essentially the same information as the $B_i$, what do we attain? I.e. why is $\hat{B}_i$ a useful representation?  For example, sec. 2.3 states "we attach certificates to extracted mechanisms", but since the extracted mechanisms are in fact the full weights of each block, it is unclear what the certificate is really showing: when would it be the case that the full weights of each block do not reproduce the activations of the original transformer?

5. Additional results on a larger model. Reviewers 6KNo and WtbH request that experiments are run on a larger model. This is provided. In general, the experimental section needs clarifying as follows:
	a. Section 6.1 Give: number of prompts, which blocks are included, what the different prompt configurations are. Results show high fidelity, but again the question arises: since the extraction mechanism simply copies all weights, why would we not expect this high fidelity?
	b. Section 6.2 Now we have a default prompt set and a stress-test configuration. Explain this in more detail. Are the blocks extracted on the default set and tested on the stress-test configuration? Why is this not followed for GPT-2? Activation coverage falls here somewhat. Is this due to numerical error as mentioned in sec 3.1 or is there another source of error? Please make it clear.

6. Baselines. Reviewer 6KNo mentions that baselines would be useful. This has been provided in the form of low-rank approximations of attention projection matrices. These results are useful and show how the pipeline can be used to provide certificates for extracted blocks that are not simply copies of the full weights.

7. Realistic application. Reviewers WtbH and 6KNo requests that the paper evaluate BlockCert in realistic datasets and tasks, and show that certificates help diagnose or bound side effects compared to baselines. The response is that the existing refusal/safety corpus provides this. However, this experiment is very small with only 24 prompts, and the reviewers are asking for a more realistic application scenario. I would therefore ask that a slightly larger (~200 prompt) experiment is carried out.

7. Content of certificates. Reviewer WtbH asks for semantic fidelity metrics in the certificates. The response is that semantic refusal/answer accuracies are reported in the edit certificates. However, this is not detailed in section 4.1. This should be included in this section, or an argument given as to why it is not needed.

8. Reviewer 6KNo asks for details on the sensitivity of the model to values of $\tau$ and $\alpha$, and also computational complexity and runtime. Complexity and runtime are reported and sensitivity to $\alpha$ is reported in Table 4. Please add a similar table reporting sensitivity to $\tau$.

As well as these content points, the paper should be improved stylistically in the following ways:

1. Reviewer 6KNo notes that the emphasis on file formats is unnecessary (JSON, `.npz`). This is really an implementation detail, and can be described in the GitHub repository.

2. In section 6, results do not need to be reported with reference to their JSON files. I.e., '`xxx.json` reports that' can be dropped. When comparable, it is better to report results in a table for ease of comparison. Implementation details such as where artifacts are stored do not need to be mentioned in the paper: this can be described in the GitHub readme, with reference back to paper sections.

3. Code is not available. Making the code available for inspection under Anonymous GitHub (https://anonymous.4open.science/) would be useful at the review stage.

**Audience:**

Yes

**Audience Explanation:**

Reviewers liked the end-to-end workflow and the use of the Lipschitz bound as a bookkeeping bridge (6KNo) and the use of Lean to formalize this as a theorem (WtbH). The method is lauded as being lightweight and practical (WtbH). Overall, reviewers also liked the placement of this paper as addressing an important gap between mechanistic interpretability and formal verification.

**Claims And Evidence:**

No

**Claims Explanation:**

This paper proposes BlockCert, a framework for certified blockwise extraction of transformer mechanisms, which extracts structured surrogate implementations for residual blocks together with machine-checkable certificates that bound approximation error, record coverage metrics, and hash the underlying artifacts for a pre-trained transformer and a prompt distribution. Certified blocks can be stitched together to provide a model-level certification, and the behaviour of the model can be edited.

Reviewers raised some key points: placement in the literature and comparison with the state-of-the-art; a change to an emphasis on tooling; testing on larger models; adding baselines; and a realistic application scenario. Some reviewers mentioned a lack of novelty: this is not a criterion for TMLR and therefore this critique has been disregarded.

The paper has been updated with additional results on a larger model (Llama-2-7B), baselines, and a realistic application scenario, but is still lacking in certain respects, and therefore cannot be accepted in its present form. Detailed comments are given in the 'Additional Comments' section.

**Resubmission Of Major Revision:**

The authors may consider submitting a major revision at a later time.